



# Improved representation of volcanic sulfur dioxide depletion in Lagrangian transport simulations: a case study with MPTRAC v2.4

Mingzhao Liu[1], Lars Hoffmann[1], Sabine Griessbach[1], Zhongyin Cai[2], Yi Heng[3,4,5], and Xue Wu[6]

[1]Jülich Supercomputing Centre, Forschungszentrum Jülich, Jülich, Germany
[2]Yunnan Key Laboratory of International Rivers and Transboundary Eco-security, Institute of International Rivers and Eco-Security, Yunnan University, Kunming, China
[3]School of Computer Science and Engineering, Sun Yat-sen University, Guangzhou, China
[4]National Supercomputing Center in Guangzhou (NSCC-GZ), Guangzhou, China
[5]Guangdong Province Key Laboratory of Computational Science, Guangzhou, China
[6]Institute of Atmospheric Physics Chinese Academy of Sciences: Beijing, Beijing, China

**Correspondence:** Mingzhao Liu (mi.liu@fz-juelich.de)

**Abstract.** The lifetime of sulfur dioxide ($SO_2$) in the Earth's atmosphere varies from orders of hours to weeks, mainly depending on whether cloud water is present or not. The volcanic eruption on Ambae Island, Vanuatu, in July 2018 injected a large amount of $SO_2$ into the upper troposphere and lower stratosphere (UT/LS) region with abundant cloud cover. In-cloud removal is therefore expected to play an important role during long-range transport and dispersion of $SO_2$. In order to better represent

the rapid decay processes of $SO_2$ observed by the Atmospheric InfraRed Sounder (AIRS) and the TROPOspheric Monitoring Instrument (TROPOMI) in Lagrangian transport simulations, we simulate the $SO_2$ decay in a more realistic manner compared to our earlier work, considering gas phase hydroxyl (OH) chemistry, aqueous phase hydrogen peroxide ($H_2O_2$) chemistry, wet deposition, and convection. The either newly developed or improved chemical and physical modules are implemented in the Lagrangian transport model Massive-Parallel Trajectory Calculations (MPTRAC) and tested in a case study for the July

2018 Ambae eruption. To access the dependencies of $SO_2$ lifetime on the complex atmospheric conditions, sensitivity tests are conducted by tuning the control parameters, changing the release height, predefined OH climatology data, the cloud pH value, the cloud cover and other. Wet deposition and aqueous phase $H_2O_2$ oxidation remarkably increased the decay rate of the $SO_2$ total mass, which leads to a rapid and more realistic depletion of the Ambae plume. The improved representation of chemical and physical of $SO_2$ loss processes described here is expected to lead to more realistic Lagrangian transport simulations of

volcanic eruption events with MPTRAC in future work.

## 1 Introduction

Volcanic eruptions have a strong impact on human living by causing various hazards and destructive impacts on human beings' living conditions. At the ground, $SO_2$ and acidic aerosols from eruption gases cause health hazards and increase respiratory morbidity and mortality (Hansell and Oppenheimer, 2004; Schmidt et al., 2011). The wet deposition of $SO_2$ leads to acid

precipitation that has a destructive effect on the ecosystem and environment, including acidifying the soil, contaminating the water sources and damaging the vegetation (Delmelle et al., 2002). In the UT/LS, sulfate aerosol formed by $SO_2$ oxidation has



a significant impact on the radiative forcing and energy balance of the Earth by scattering solar radiation and by absorbing and re-emitting thermal emissions (Robock, 2000; Kloss et al., 2020; Malinina et al., 2021). Satellite observations and atmospheric chemistry-transport modelling allow us to monitor the transport of volcanic ash and $SO_2$ and to better plan for evacuation

and hazard mitigation. Studies on $SO_2$ long-range transport and dispersion can also help to better understand atmospheric dynamics, for instance the impact of the Asian monsoon on aerosol transport (Wu et al., 2017).

Remote sensing observations from satellite instruments are widely used assets to monitor and study volcanic activity. Satellite observations provide high-resolution $SO_2$ measurements on a global scale, which are particularly useful to initialize and evaluate the results of Lagrangian transport simulations. Among these instruments, the Atmospheric InfraRed Sounder (AIRS)

(Aumann et al., 2003) and the TROPOspheric Monitoring Instrument (TROPOMI) (Veefkind et al., 2012) both provide data products that are capable of detecting volcanically enhanced $SO_2$ concentrations in the UT/LS. Hoffmann et al. (2014) proposed an $SO_2$ index based on brightness temperature difference for AIRS observations, which is most sensitive to $SO_2$ layers at 8 to 13 km and well suited to detect volcanic emission. The TROPOMI Level-2 $SO_2$ product provides volcanic $SO_2$ total column amounts for prescribed $SO_2$ plume heights at 1, 7, and 15 km with high horizontal resolution (Theys et al., 2020).

These data products provide valuable references to evaluate and improve our transport model in the present study.

For the simulation of volcanic $SO_2$ transport, Lagrangian particle dispersion models are important tools that can resolve both small-scale features and long-range transport by calculating trajectories of ensembles of air parcels driven by deterministic (wind field and buoyancy) and stochastic (turbulence and convection) dynamics. In our past research, the Massive-Parallel Trajectory Calculations (MPTRAC) model (Hoffmann et al., 2016, 2022a) has been utilized to study volcanic eruption events,

including research on source reconstruction (Heng et al., 2016; Cai et al., 2022), long-range transport (Wu et al., 2017, 2018), and large-scale parallel inverse modelling (Liu et al., 2020). Most of these studies predefined the $SO_2$ lifetime as a constant empirical value. However, the lifetime of $SO_2$ varies from orders of hours to weeks, depending on whether liquid or ice clouds are present or not (Eatough et al., 1994; McGonigle et al., 2004; Khokhar et al., 2005). In the gas phase, $SO_2$ is mainly depleted by reaction with OH, whereas in the presence of clouds $SO_2$ can be dissolved in cloud droplets and removed by precipitation

or aqueous phase oxidation. Due to the variability of the complicated atmospheric background conditions, it is hard to use a specific number to represent the local $SO_2$ residence time. Therefore, we aim to model the deposition of $SO_2$ with a more realistic representation of physical and chemical processes in the framework of MPTRAC, including oxidation in the gas and aqueous phase and wet deposition in this paper.

Ambae Island (15.39°S, 167.84°E), located in the South Pacific in Vanuatu, contributed the largest volcanic eruption in the

year 2018. Among four main eruption phases during 2017 and 2018, the most intensive one in July 2018 injected at least $4 \times 10^8$ kg of $SO_2$ to a peak altitude of ∼17 km (Moussallam et al., 2019). The volcanic $SO_2$ injected into the UT/LS formed aerosol particles that have a significant impact on atmospheric radiative forcing and global climate (Kloss et al., 2020; Malinina et al., 2021). Compared with other cases, e. g., the Raikoke eruption in 2019 (Cai et al., 2022; de Leeuw et al., 2021), Kasatochi in August 2008, Sarychev in June 2009 and Nabro in June 2011 (Höpfner et al., 2015) with $SO_2$ lifetimes of about 14, 13, 24

and 32 days, respectively, the Ambae case in July 2018 has a much shorter lifetime of ∼ 4 days (Malinina et al., 2021). Local reports of acid rain suggest that the eruption was accompanied by strong wet deposition, which means that the released $SO_2$





encountered significant wet removal. As we aim to better understand and represent these processes in the MPTRAC model, we selected the Ambae eruption in July 2018 as a case study for this work.

The paper is organized as follows. Section 2 introduces the updates on the chemical and physical modules in MPTRAC and provides a brief introduction to the AIRS and TROPOMI satellite observations. In Sect. 3, simulation results for the Ambae eruption in July 2018 are presented and evaluated, including the baseline simulation and various parameter sensitivity tests on the OH chemistry module, the $H_2O_2$ chemistry module, the wet deposition module, and the convection module. Section 4 provides the summary and conclusions of the study.

## 2 Data and methods

### 2.1 The MPTRAC Lagrangian transport model

For the simulation of the dispersion and depletion of $SO_2$ from the Ambae eruption, we applied the Massive-Parallel Trajectory Calculations (MPTRAC) model (Hoffmann et al., 2016, 2022a). Mainly, trajectories of air parcels are calculated by given horizontal winds and vertical velocities of meteorological input data, with additional stochastic perturbations being added to simulate diffusion and subgrid-scale wind fluctuations. Additionally, several improved or newly developed chemical and physical modules of MPTRAC are applied in the simulations. In particular, the $SO_2$ mass of the air parcels is decomposed by OH oxidation and wet removal processes instead of simply using an exponential decay with a fixed lifetime as applied in our earlier studies. The new and revised modules are described in the following sections.

The transport simulations with MPTRAC are driven by the European Centre for Medium-Range Weather Forecasts' (ECMWF's) fifth generation reanalysis ERA5 (Hersbach et al., 2020) with hourly meteorological data at $0.3° \times 0.3°$ horizontal resolution on 137 vertical levels. This is a significant improvement in spatiotemporal resolution compared with the previous generation ERA-Interim reanalysis (Dee et al., 2011), providing only $0.75° \times 0.75°$ horizontal resolution on 60 vertical levels at 6-hourly time intervals. The higher resolution ERA5 data are expected to lead to more accurate Lagrangian transport simulations compared to ERA-Interim (Hoffmann et al., 2019).

MPTRAC is a Lagrangian transport model, which is developed with a hybrid MPI-OpenMP-OpenACC parallelization scheme for application on CPU/GPU heterogeneous supercomputers, aiming for good parallel performance and scaling efficiency. The computing tasks in MPTRAC are distributed over the compute nodes and compute cores by MPI and OpenMP and offloaded to GPUs by means of OpenACC for faster and more energy-efficient computational performance (Liu et al., 2020; Hoffmann et al., 2022a). In this study, each individual simulation is conducted in parallel with 48 OpenMP threads. MPI parallel multi-processing for different model parameter settings is applied in the sensitivity tests, which significantly improves the overall runtime of the simulations. On the state-of-the-art high performance computing system Jülich Wizard for European Leadership Science (JUWELS) at the Jülich Supercomputing Centre (Jülich Supercomputing Centre, 2019), individual Lagrangian transport simulations with $10^6$ air parcels over a time period of 15 days require about 1.5 h of total runtime on a compute node. GPU acceleration was not considered here as there is no increase in computational speed at this problem size.



### 2.1.1 Hydroxyl radical oxidation in the gas phase

The hydroxyl radical (OH) is an important oxidant in the atmosphere, which causes rapid decay of many gas phase species. In MPTRAC, the mass decay of a trace gas of an air parcel is described by an exponential formula,

$$m(t + \Delta t) = m(t) \exp(-k\Delta t).$$ (1)

Assuming that the concentration of OH is in near steady-state, the pseudo first-order reaction coefficient $k$ is calculated as

$$k = k_f \times [\text{OH}]$$ (2)

with an effective second-order coefficient $k_f$ and a prescribed monthly zonal mean OH field. The oxidation of $SO_2$ with OH is a termolecular reaction,

$$\text{SO}_2 + \text{OH} \rightarrow [\text{HOSO}_2]^* \xrightarrow{M} \text{HOSO}_2,$$ (3)

where the excited intermediate $[\text{HOSO}_2]^*$ requires an inert molecule M (e. g., $N_2$ or $O_2$) to remove the energy and stabilize it into sulfate. In the high-pressure limit, the rate-limiting step is the production of $[\text{HOSO}_2]^*$, while in the low-pressure limit, the

reaction rate depends on the abundance of M and the production of $HOSO_2$. Thus, the effective second-order rate coefficient of the $SO_2$-OH oxidation process is temperature- and pressure-dependent, which is described here by using a formula given by the NASA Jet Propulsion Laboratory (JPL) data evaluation (Burkholder et al., 2019) as

$$k_f(T, [M]) = \frac{k_0(T) [M]}{1 + \frac{k_0(T) [M]}{k_\infty(T)}} 0.6^{\left\{ 1 + \left[ \log_{10} \left( \frac{k_0(T) [M]}{k_\infty(T)} \right) \right]^2 \right\}^{-1}}.$$ (4)

The high-pressure limit rate $k_\infty = 2.9 \times 10^{-31} \times (\frac{T}{298})^{-4.1} \, \text{cm}^3 \, \text{molecule}^{-1} \, \text{s}^{-1}$ and the low-pressure limit rate $k_0 = 1.7 \times$

$10^{-12} \times (\frac{T}{298})^{0.2} \, \text{cm}^6 \, \text{molecule}^{-2} \, \text{s}^{-1}$ were also obtained from the JPL evaluation.

As for the prescribed zonal mean OH fields, monthly mean climatology of OH calculated from simulation of the Chemical Lagrangian Model of the Stratosphere (CLaMS) is used by default (Pommrich et al., 2014). Another OH data set considered in this study was obtained from integrated the Copernicus Atmosphere Monitoring Service (CAMS) reanalysis OH data into monthly latitude- and pressure-dependent fields, keeping consistency with the CLaMS climatology(Inness et al., 2019). The

CLaMS data has 18 latitude bins and 34 pressure levels while CAMS data has 241 latitude bins and 25 pressure levels. CLaMS has a complete chemistry scheme on the stratospheric chemistry while the CAMS reanalysis uses a chemical mechanism most suitable for the troposphere. A comparison of the CLaMS and CAMS data sets with in-situ measurements is presented in the appendix. Whether the difference between the two OH datasets affects the MPTRAC simulations for the Ambae case study is discussed.

The formation of OH is driven by the photolysis of ozone in the troposphere and $H_2O$ in the stratosphere, which causes a strong correlation with diurnal variability (Minschwaner et al., 2011). In contrast to previous versions of MPTRAC, which did not take diurnal variability into account, the mean OH climatology concentration $[\text{OH}]_0$ is multiplied here by a scaling factor





depending on the solar zenith angle $\theta_{\text{SZA}}$ as proposed by Minschwaner et al. (2011) to model the diurnal variations,

$$f(\theta_{\text{SZA}}) = \exp[-\beta \sec(\theta_{\text{SZA}})]. \tag{5}$$

The term $\sec(\theta_{\text{SZA}})$ is the approximate air mass factor that represents the ratio of the optical slant path to the effective vertical path. The parameter $\beta$ represents the vertical optical depth. Based on Minschwaner et al. (2011), a $\beta$ value of 0.6 is used for simulations covering the UT/LS region. To maintain the same mean values of the scaled data as the monthly OH field, it is divided by a normalization factor that is obtained by integrating the correlation factor $f$ over longitude $\lambda$. The final OH concentration is calculated as

$$[\text{OH}] = \frac{[\text{OH}]_0 f(\theta_{\text{SZA}})}{\int_{-180}^{180} f(\theta_{\text{SZA}}(\lambda))d\lambda/360}. \tag{6}$$

    Figure 1 shows average vertical profiles of the different OH data sets at tropical latitudes (from 23.5°S to 23.5°N). The CLaMS OH data (Pommrich et al., 2014), calculated using methane and ozone data taken from the HALOE climatology for the year 2005 (Grooß and Russell III, 2005), are compared with the CAMS OH reanalysis data for the years 2005 and 2018, respectively. The differences in the CAMS mean profiles between 2005 and 2018 are found to be negligible. Comparing the

CLaMS and CAMS OH data, these two datasets show similar concentrations in the troposphere while in the stratosphere, CLaMS OH concentrations are much larger than CAMS OH concentrations. To further evaluate the OH fields of the climatologies, we compared them to several NASA in-situ OH measurements in the troposphere at similar altitudes, solar zenith angles, and time periods and also to Microwave Limb Sounder(MLS) satellite data in the stratosphere. Both, CLaMS and CAMS data showed good agreement with the in-situ measurements in the troposphere. At altitudes above 20 km, CLaMS data show much

better agreement with MLS observations than the CAMS reanalysis. Further details of the comparison are presented in the appendix.

### 2.1.2   Hydrogen peroxide oxidation in the aqueous phase

The in-cloud oxidation pathway of $SO_2$ is mainly dominated by the reaction with hydrogen peroxide ($H_2O_2$),

$$SO_2 + H_2O \rightleftharpoons H^+ + HSO_3^-$$
$$HSO_3^- + H_2O_2 \rightarrow SO_4^{2-} + H^+ + H_2O \tag{7}$$

Another oxide of $SO_2$ in liquid phase, ozone, is not considered here because its effects are expected to be negligible when $pH \leq 5$ (Rolph et al., 1992; Pattantyus et al., 2018). In contrast to OH, $H_2O_2$ has a longer atmospheric lifetime and cannot be treated as a static field as in the OH chemistry module. The reaction will rapidly deplete $H_2O_2$ in a few minutes (Pattantyus et al., 2018; Redington et al., 2009), keeping the $H_2O_2$ concentration in the aqueous phase much lower than equilibrium conditions from Henry's Law (Barth et al., 1989). To approximate the concentration of $H_2O_2$ in the aqueous phase, we use an

expression following Rolph et al. (1992) that is approximated from measurement data in Barth et al. (1989),

$$[H_2O_2]_{aq} = H_{H_2O_2} \times [H_2O_2]_g \times 0.59 e^{-0.687[SO_2]_{vmr}} \tag{8}$$



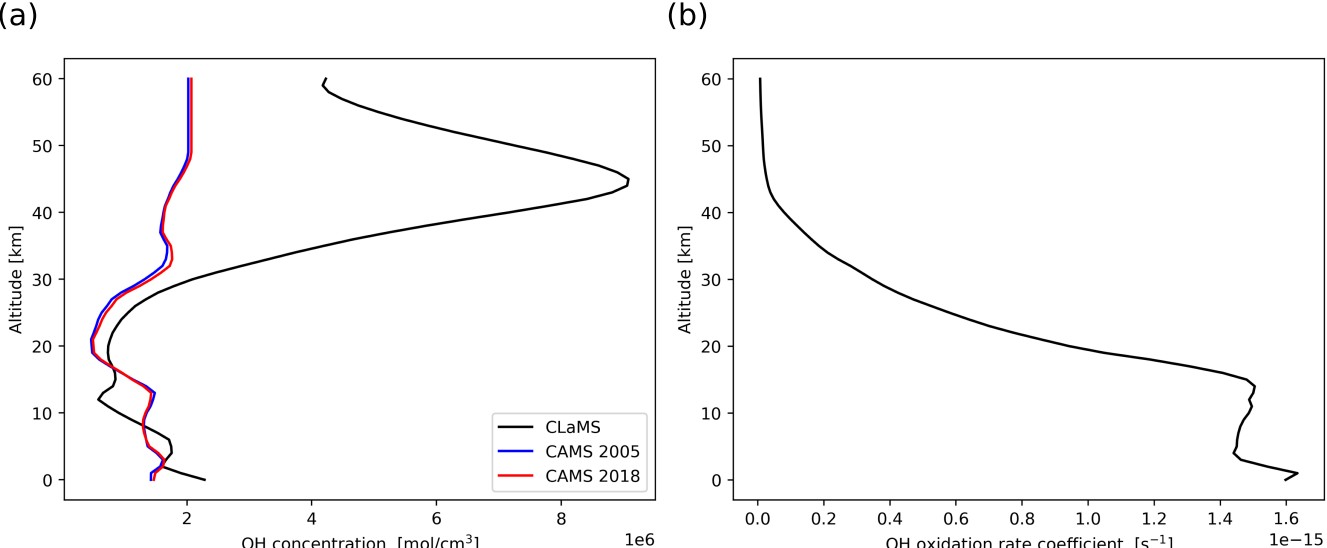

**Figure 1.** Vertical profiles of (a) OH concentrations over the tropics retrieved from CLaMS and CAMS climatology data, respectively, on 26 July 2018, 00:00 UTC and (b) OH oxidation rate coefficients at the same time.

Here, $H_{\mathrm{H_2O_2}}$ represents the Henry's law constant of $H_2O_2$ (Sander, 2015). The concentration of $H_2O_2$ in the atmosphere is defined using monthly mean zonal mean data extracted from the CAMS reanalysis. The volume mixing ratio of $SO_2$ (in units of ppbv) is calculated in grid boxes of $1° \times 1°$ in the horizontal and $0.2\,\mathrm{km}$ in the vertical direction.

The concentration of the $SO_2$ in the aqueous phase is converted into a mass concentration in the air by multiplying the cloud water volume content $L$, thus the oxidation rate of $SO_2$ by $H_2O_2$ is formulated as (Rolph et al., 1992):

$$\frac{d[SO_2]_{aq}}{dt} = \frac{d[SO_2]_g/L}{dt} = -k_{\mathrm{H_2O_2}}[H_2O_2]_{aq}K'H_{SO_2}[SO_2]_g. \tag{9}$$

Here, $K'$ is the dissociation constant of $H_2SO_3$ (Berglen et al., 2004) and $L$ is the volume liquid water content of the clouds. Combining Eqs. (8) to (9), the rate coefficient used in Eq. (1) of the $H_2O_2$ aqueous phase chemistry module is formulated as

$k = k_{\mathrm{H_2O_2}}[H_2O_2]_{aq}K'LH_{SO_2}. \tag{10}$

The reaction rate coefficient $k_{\mathrm{H_2O_2}}$ is formulated as (Maass et al., 1999)

$$k_{\mathrm{H_2O_2}} = 9.1 \times 10^7 \times \exp\left[29700/R \times \left(\frac{1}{T} - \frac{1}{298.15}\right)\right]. \tag{11}$$

When the air parcel is located inside a grid box where the cloud water content is non-zero, the $H_2O_2$ oxidation scheme is activated to contribute to the depletion of $SO_2$.

In reanalysis data, cloud information always has uncertainties and cannot resolve finer scale cloud structures, which means that even in a cloudy grid box, the in-cloud oxidation may locally not take effect. Under this aspect, we implemented a control





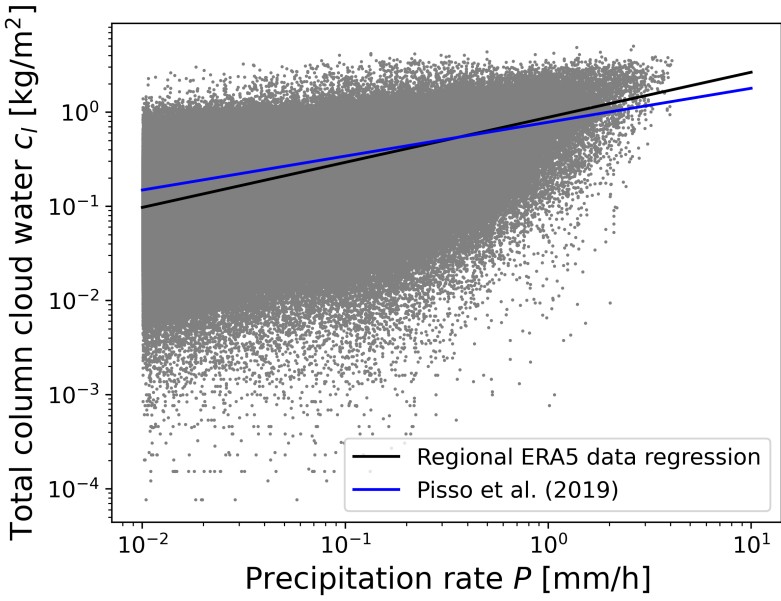

**Figure 2.** Scatter plot of total column cloud water versus precipitation rate of ERA5 meteorological data for a region and time period covering the Ambae eruption (see text for details). The black line represents the regression of the data as given in Eq. (12). The blue line shows the expression given by Pisso et al. (2019).

parameter allowing us to modify the cloud cover and to decrease the fraction of in-cloud oxidation. Only when a random number in the range of 0 to 1 is smaller than the given control parameter, the $H_2O_2$ chemistry will be activated. At present, the cloud cover default value is given as 0.8. In future versions of MPTRAC, the meteorological cloud cover can be used instead

of a pre-defined value.

### 2.1.3 Wet deposition

Irreversible in-cloud and below-cloud wet removal processes of trace gases need to be treated separately. In MPTRAC, the cloud liquid water content (CLWC) and the cloud ice water content (CIWC) of the meteorological input data are used to determine whether an air parcel is located within or below a cloud. The total column cloud water $c_l$ as well as the cloud depth

$Z$ are additional input variables to the wet deposition module. Similar to Pisso et al. (2019), the precipitation rate $P$ (in units of mm h$^{-1}$) is estimated from $c_l$ (in units of kg m$^{-2}$) by means of a regression analysis using the ERA5 meteorological data for a region covering the Ambae eruption (10°S to 30°N, 160°W to 140°E) over the time period from 25 July to 31 July in 2018, as shown in Fig. 2,

$$c_l = 0.763 \times P^{0.478}. \tag{12}$$

For the modeling of in-cloud wet deposition, a similar exponential removal as in Eq. (1) is used with a scavenging coefficient $\Lambda$ (in units of $s^{-1}$). The in-cloud wet deposition process is implemented with two schemes for a choice. The first scheme is



based on the rain-out rate of cloud water, which determines the removal of a soluble trace gas taken up by cloud droplets and removal by precipitation according to the solubility of the trace gas. The in-cloud scavenging coefficient is calculated by multiplying the partition ratio of the species in the aqueous phase versus the gas phase $\alpha$ with the cloud water removal rate

(Slinn, 1974; Levine and Schwartz, 1982; Garrett et al., 2006),

$$\Lambda = \alpha \frac{P}{LZ}, \tag{13}$$

where $L$ is the volume liquid water content of the cloud (in units of $m^3 \cdot m^{-3}$), $Z$ is the depth of the cloud layer determined by the pressure of cloud top and cloud bottom as taken from the meteorological input data, and $\alpha$ is defined by

$$\alpha = \frac{N_l}{N_g} = \frac{HP_xL}{P_x/\mathrm{R}T}, \tag{14}$$

according to the ideal gas law, where $H$ is Henry's law coefficient, R is the universal gas constant, and $P_x$ is the partial pressure of species $x$.

Combining Eqs. (13) and (14), a formula similar to the approach of the HYSPLIT model (Draxler and Hess, 1998) is obtained,

$$\Lambda = \eta H \mathrm{R} T P Z^{-1}. \tag{15}$$

The factor $\eta$ represents a temperature dependent retention coefficient in cloud versus equilibrium concentration in liquid water. The default value of $\eta$ is set following Webster and Thomson (2014), assuming the cloud between 238.15 K to 273.15 K to be in the mixed phase, and a retention ratio in ice clouds to be 0.15:

$$\eta = \begin{cases} 1, & \text{if } T \geq 273.15\,\mathrm{K} \\ 0.15, & \text{if } T \leq 238.15\,\mathrm{K} \\ 0.15 + \frac{T - 238.15\,\mathrm{K}}{273.15\,\mathrm{K} - 238.15\,\mathrm{K}}(1 - 0.15), & \text{if } 238.15\,\mathrm{K} < T < 273.15\,\mathrm{K} \end{cases} \tag{16}$$

$SO_2$ is a moderate soluble gas with a Henry's Law constant of $1.3\,\mathrm{M\,atm^{-1}}$ at 298 K (Sander, 2015). However, the solubility

of $SO_2$ strongly depends on the pH value because it undergoes dissociation. In the simulations for $SO_2$, the partition of $SO_2$ in clouds is represented by an effective Henry's law constant related to the pH value to account for the dissolution and dissociation of $SO_2$ in cloud water (Berglen et al., 2004),

$$H_{\mathrm{eff}}(SO_2) = H(SO_2) \times \left[1 + K'/\left[H^+\right] + K'K''/\left[\mathrm{H}^+\right]^2\right]. \tag{17}$$

Here $K'$ and $K''$ are the first and second dissociation constants of $SO_2$, using the formulation of Berglen et al. (2004), and

$H(SO_2)$ is the Henry's law constant (Sander, 2015). With this method, the pH value becomes a control parameter of the model affecting the wet deposition, for which we assumed a default value of 4.5, guided by earlier work (Berglen et al., 2004; Koch et al., 1999).

For comparison, we also implemented another in-cloud wet deposition scheme applying an empirical exponential expression given by

$$\Lambda = \eta a P^b. \tag{18}$$



The choice of the parameters $a$ and $b$ follows the UK Met Office's Numerical Atmospheric-dispersion Modelling Environment (NAME) (Webster and Thomson, 2014).

The below-cloud wet deposition is a washout process through impact or diffusion with raindrops. With respect to $SO_2$, the washout rate has a typical magnitude of $\sim 10^{-5} s^{-1}$ (Maul, 1978; Martin, 1984; Elperin et al., 2015). Here, we set the

parameters for $SO_2$ to $a = 2 \times 10^{-5}$ and $b = 0.616$, which follows the settings in the FLEXPART model (Pisso et al., 2019).

### 2.1.4 Convection

Due to the limited resolution of the global meteorological input data, neither ERA-Interim nor ERA5 are capable of resolving subgrid-scale convection processes. In MPTRAC, the extreme convection parametrization (Draxler and Hess, 1998; Gerbig et al., 2003) is used to represent the effects of convective up- and downdrafts being unresolved in the meteorological input data.

The lifetime and depletion of $SO_2$ typically have a strong dependency on the atmospheric conditions at different altitudes. To test the impact of parametrized convection on the $SO_2$ transport simulations is part of the sensitivity tests presented this work.

The extreme convection parametrization requires the convective available potential energy (CAPE) and the height of the equilibrium level (EL) for input. CAPE represents the vertical atmospheric instability by integrating the local buoyancy of an air parcel from the level of free convection (LFC) to the EL,

$$CAPE = \int_{z_{LFC}}^{z_{EL}} g \left( \frac{T_{v,ap} - T_{v,env}}{T_{v,env}} \right) dz, \tag{19}$$

where $T_{v,ap}$ is the virtual temperature of the air parcel and $T_{v,env}$ is the virtual temperature of the environment. If the CAPE value is larger than a given threshold $CAPE_0$, an air parcel will be randomly redistributed between the surface and the equilibrium level, weighted by density. The method is similarly handled in the convective transport scheme in the Stochastic Time-Inverted Lagrangian Transport (STILT) model (Gerbig et al., 2003). A more detailed description and discussion of the

calculation of the CAPE and EL values in MPTRAC is given by Hoffmann et al. (2022a).

## 2.2 Satellite data products

### 2.2.1 AIRS sulfur dioxide measurements

The Atmospheric InfraRed Sounder (AIRS) (Aumann et al., 2003) is an infrared spectrometer aboard the National Aeronautics and Space Administration's (NASA's) Aqua satellite. Aqua operates in a sun-synchronous low Earth orbit at orbit altitude of

705 km, providing nearly continuous measurements since September 2002. The AIRS instrument has across-track scanning capabilities. The swath width is 1780 km consisting of 90 footprints per scan with a footprint size of 13.5 km $\times$ 13.5 km at nadir. The measurements take place at about 01:30 and 13:30 local time for the descending and ascending sections of the orbits, respectively.

To detect the presence of volcanic $SO_2$ using the AIRS radiance measurements, an $SO_2$ index defined as the brightness

temperature difference between 1407.2 cm$^{-1}$ and 1371.5 cm$^{-1}$ in the 7.3 $\mu$m $SO_2$ waveband is used (Hoffmann et al., 2014).





The $SO_2$ index of Hoffmann et al. (2014) is most sensitive in the column density range of about 10 to 200 Dobson Units (DU) at altitudes of 8 to 13 km, which covers explosive volcanic eruptions with $SO_2$ injections into the UT/LS region. For $SO_2$ index values larger than 4 K, the $SO_2$ index is able to clearly detect volcanic plumes.

### 2.2.2 TROPOMI sulfur dioxide measurements

The TROPOspheric Monitoring Instrument (TROPOMI) (Veefkind et al., 2012) aboard the European Space Agency's Sentinel-5 Precursor satellite in a near-polar sun-synchronous orbit measures ultraviolet, visible, near-infrared, and shortwave infrared spectra at daytime. The TROPOMI instrument provides high spatial resolution with a pixel size of 7 km×3.5 km over a swath width of 2600 km.

The TROPOMI products include volcanic $SO_2$ abundances for prescribed plume heights of 1, 7, and 15 km, based on a
detection algorithm described by Brenot et al. (2014) that recognizes enhanced $SO_2$ values from volcanic eruptions as well as anthropogenic sources. In this study, we use the TROPOMI Level 2 product with a priori $SO_2$ profiles centered around 15 km (Theys et al., 2017) for the time period of 24 July to 11 August 2018 during the Ambae eruption.

The $SO_2$ total mass of the Ambae eruption was calculated using the TROPOMI data to compare with the simulation results. TROPOMI scans the Ambae region at around 00:00 UTC. A grid with resolution of 0.1° covering the longitude range from
120°E to 110°W and the latitude range from 50°S to 30°N was used to calculate the average mass of $SO_2$ in each grid box and to sum up the total mass. A filter of $\theta_{SZA} < 70°$ and the detection flag to filter out anthropogenic $SO_2$ were used for the analysis of the TROPOMI volcanic $SO_2$ data product as described in (Theys et al., 2020). The derived $SO_2$ total mass curve is very similar to the result of Malinina et al. (2021).

## 3 Results

### 255 3.1 Baseline simulation

In this section, we present a baseline simulation of the dispersion and depletion of the volcanic $SO_2$ plume of the Ambae eruption in July 2018. In our initial tests, it was found that with only OH oxidation and wet deposition being considered, the simulations cannot fully explain the observed fast depletion of $SO_2$. Therefore, we newly implemented the process of in-cloud oxidation with $H_2O_2$. As the plume transport in this case occurs in a high-CAPE tropical region, the convection module
was also included. A CAPE threshold of 1000 J kg$^{-1}$ was used to include moderate to strong subgrid-scale convection in the baseline simulation. All the parameter settings in the baseline simulation use the default values as introduced in Sect. 2. In the following sections, we will analyze the sensitivity of parameter choices for the above mentioned chemistry and physics modules of MPTRAC with respect to the baseline simulation setup.

Figure 3a shows the $SO_2$ total mass curve calculated from the TROPOMI data. The total mass of $SO_2$ in the UT/LS region
shows a strong increase on 26 July 2018, reaching a peak on 28 July, and decreases back to pre-eruption levels on 7 August 2018. According to different satellite instrument observations of the Support to Aviation Control Service (SACS, 2022), the



volcanic plume is detected directly over the volcano from 26 to 27 July. The bulletin report of the Global Volcanism Program (GVP) (Krippner and Venzke, 2019) states that the main explosion occurred on 26 July and another two intense episodes, producing volcanic lightning, occurred on 27 July. Based on the different satellite measurements, observational reports as well

as empirical testing, we initialized the MPTRAC baseline simulation for the Ambae case study by releasing $10^6$ air parcels with a total mass of 0.45 Tg starting on 26 July, 00:00 UTC over the time period of 36 h, assuming a Gaussian vertical profile with the maximum centered at 14 km and a full width at half maximum (FWHM) of 4 km. Although there are more advanced methods available to accurately estimate the timing and vertical distribution of volcanic emissions (Hoffmann et al., 2016; Wu et al., 2017; Cai et al., 2022), we did not apply these techniques in this study because these techniques do not fully account for $SO_2$

chemistry. This study focuses on testing the new and revised chemical and physical modules and on conducting sensitivity tests with respect to the parameter choices. As discussed below, the baseline simulation with constant emission rate and Gaussian vertical profile represents the Ambae case reasonably well so that meaningful testing and evaluation can be conducted.

To properly assess the total mass evolution and the $SO_2$ plume patterns of the baseline simulation, the model output was sampled at the exact time and location as the TROPOMI satellite footprints and output the column density of all parcels

located within a horizontal search radius of 7 km, as introduced in Hoffmann et al. (2022a). Therefore, the sample output of the model can be directly compared and analyzed with the same perspective as the TROPOMI satellite observations. Air parcels in data gaps of the satellite observations will also be excluded in the evaluation of the simulation results. A lower threshold of 1.5 Dobson units (DU) was applied to both the simulation data and the TROPOMI observations. As shown in Fig. 3a, the MPTRAC baseline simulation yields good agreement with the total mass curve derived from the TROPOMI data. The overall

$SO_2$ lifetime from the simulation is quite similar to the observational data.

Figure 3b shows the relative mass loss with respect to the total emissions over time with regard to each module. At the end of the simulation, OH chemistry, $H_2O_2$ chemistry and wet deposition lead to a mass loss of 46%, 14%, and 26% of the total mass burden, respectively. The remaining mass loss in Fig. 3a is due to air parcels leaving the study region or having $SO_2$ mass below the filtering threshold so that they are not accounted for in the mass budget anymore. The loss due to OH oxidation is

a step-shaped curve due to the diurnal variations of the OH concentration, leading to a faster loss rate at daytime and nearly zero loss at nighttime. The loss rates due to in-cloud removal processes, including wet deposition and the aqueous phase $H_2O_2$ oxidation, strongly depend on the location of the volcanic plume with respect to the clouds.

Figure 4 shows comparisons of horizontal maps of the $SO_2$ plume from the baseline simulation with AIRS and TROPOMI measurements, respectively. The bulk of the Ambae plume moved eastwards, and the moving speed is faster at lower altitudes.

A small plume at heights of 10 to 12 km moved northwards after 28 July at 160°W to 180°W, encountering strong wet deposition. The model results match the satellite observations qualitatively well with similar location and shape of the plume. At the end of the simulation the part of the plume under the cloud top has been almost depleted and the remaining part of the plume above the cloud top somewhat shows deviations from the observations, which is attributed to unpolished temporal and vertical variations of the emission estimates. Nevertheless, we consider the baseline simulation in its present form to be

suitable for further evaluation and sensitivity tests.





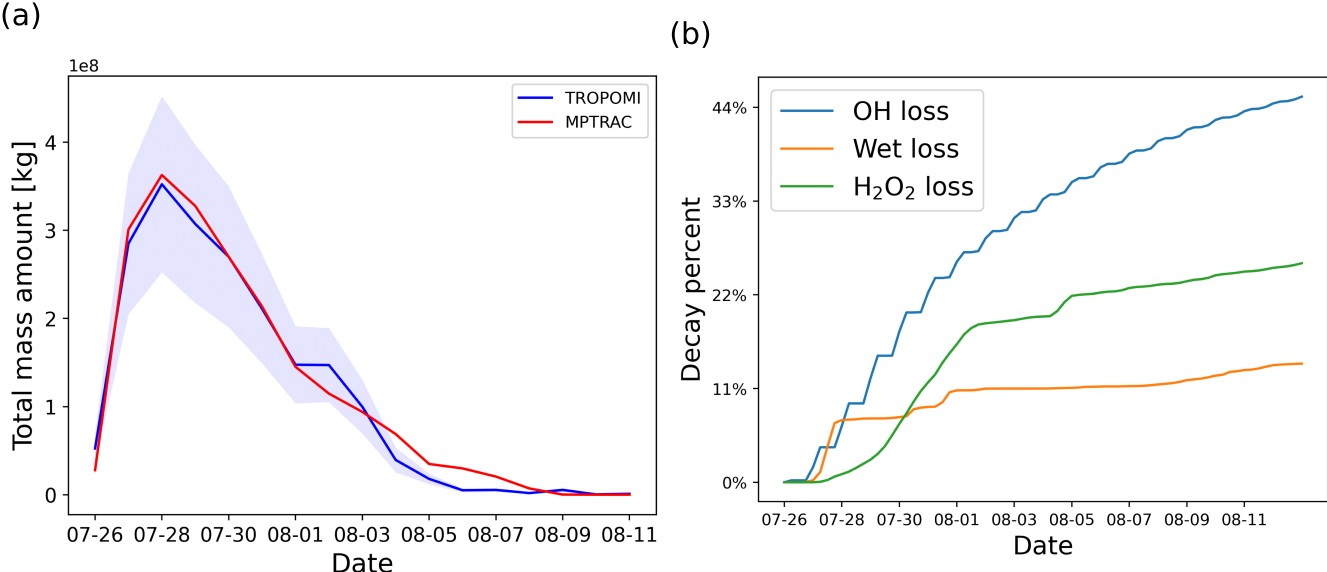

**Figure 3.** Observed (TROPOMI) and simulated (MPTRAC) $SO_2$ total mass curve (a) and mass loss (b) of the July 2018 Ambae eruption as a function of time. The blue shading represents the total error of the TROPOMI observations, combining random and systematic error components.

## 3.2 Sensitivity test on $SO_2$ release height

In this section, a series of simulations with air parcels released at different altitudes is presented to show the sensitivity of the plume injection height on the evolution of $SO_2$ total mass burden. The tests use different release heights centered at 11, 13, 15, 17, and 19 km with 2 km wide uniform vertical distribution. The convection module was not activated in this test to avoid vertical mixing of air parcels down to the surface during the simulations. As shown in Fig. 5, the $SO_2$ decay rate has an obvious dependency on release height. Releasing the volcanic $SO_2$ at different heights leads to rather different horizontal spread and different mass loss due to wet deposition, depending on whether the $SO_2$ plume encounters cloud regions or not.

As shown in Fig. 5b, the mass loss due to wet deposition shows particularly high sensitivity to the release height. Wet deposition occurs mainly on 27 to 28 July when releasing air parcels at altitudes below 14 km, while most of the wet deposition mass loss occurs around 1 August when releasing above 14 km. Almost no wet deposition occurs above 18 km because it is above the tropopause and the maximum cloud top height. Aqueous phase $H_2O_2$ chemistry is most active at below 14 km, and has a strong sensitivity on release height, while above 14 km the $H_2O_2$ chemistry is very weak. The OH gas phase oxidation does not show a clear correlation with release height, with contributions to mass loss ranging from 30% to 60% of the total mass during the simulation. However, note that the OH concentrations and temperature- and pressure-dependent rate coefficients of the OH oxidation are not linearly increasing with altitude (compare Fig. 1) over heights from 10 to 20 km. Above 15 km, the





**Figure 4.** Evolution of the Ambae SO$_2$ plume in AIRS SO$_2$ observations (a) as well as column densities from TROPOMI observations (b) and MPTRAC simulations (c). The time of the satellite observations shown here was restricted to $\pm 3$ h around 00:00 UTC. The black triangle shows the location of Ambae island. MPTRAC simulation results have been sampled on the TROPOMI footprints.

OH oxidation rate decreases with height, while below 15 km, the decay rate is controlled mainly by in-cloud removal and the OH oxidation rate quickly decreases.

Figures 6 and 7 show the cloud top height and CAPE distributions derived from the ERA5 meteorological data as well as the particles released at different altitudes, clearly indicating which part of the plume is affected by wet removal and sub-grid convection, as distinguished by yellow and blue color. The plume released at the location of Ambae Island is transported to a region with high CAPE values and abundant clouds. The wet removal and subgrid-scale convection mainly take effect at altitudes below 15 km. The maximum CAPE values range approximately from 1000 to 1600 J kg$^{-1}$. In simulations with higher injection altitudes, the air parcels are transported mainly above the cloud top and are barely influenced by wet deposition and convection.





**Figure 5.** SO$_2$ mass loss with respect to the total mass (in percent) due to OH chemistry (a), wet deposition (b), and H$_2$O$_2$ chemistry (c) as well as total mass curves (d) derived from simulations that have different release heights centered at 11, 13, 15, 17 and 19 km.

### 3.3 Sensitivity tests on OH chemistry

The SO$_2$ decay rate in the OH chemistry module is calculated by Eq. (2), which considers the temperature- and pressure-dependent reaction rate $k_f$ and the OH concentration. The OH concentration is obtained from predefined monthly mean zonal mean data and the solar zenith angle correction to account for day- and nighttime conditions. The OH data used by default in MPTRAC is the monthly mean zonal mean climatology of Pommrich et al. (2014), which was calculated using the CLaMS model chemistry scheme. For another alternative, the CAMS global reanalysis provides 3-hourly OH data with a resolution





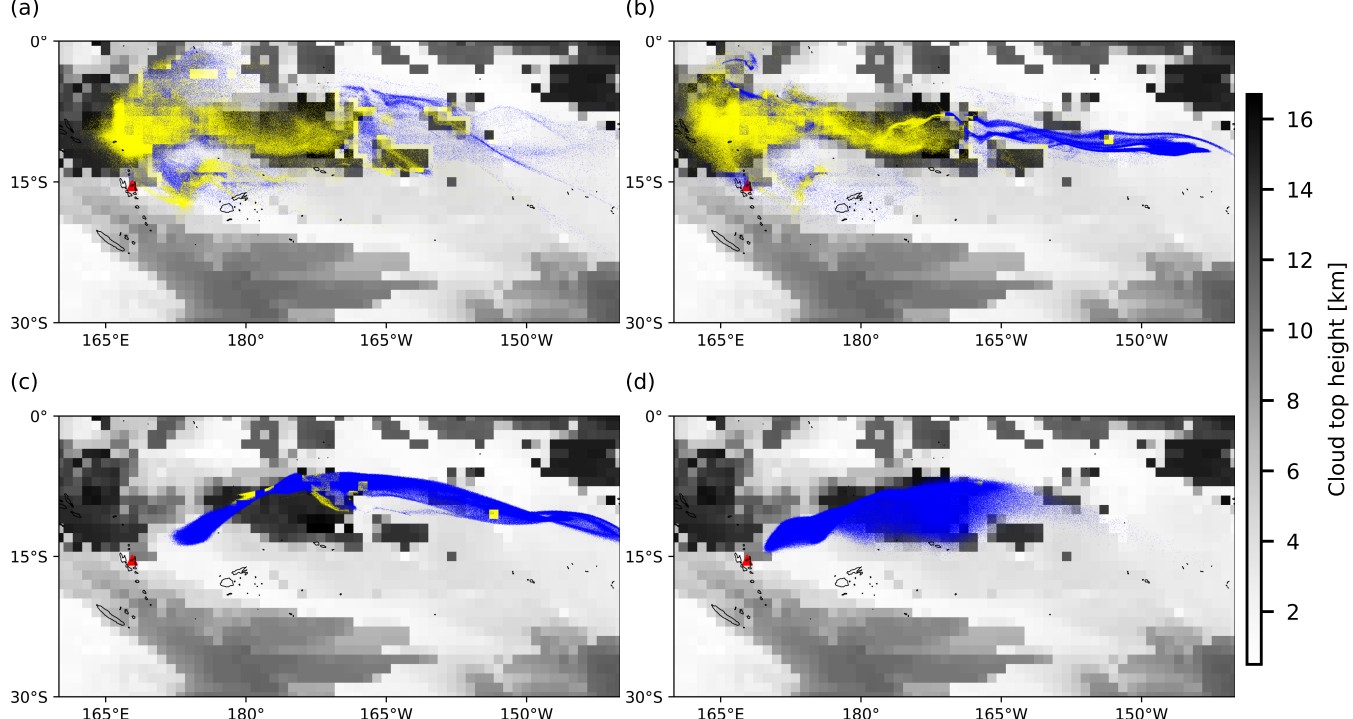

**Figure 6.** Background images represent the distributions of ERA5 cloud top height on 31 July 2018, 00:00 UTC. Dots represent air parcels that are released at altitudes of (a) 11 km, (b) 13 km, (c) 15 km, and (d) 17 km. A yellow dot indicates that the air parcel is below the cloud top height, triggering the wet deposition module, while a blue dot represents the opposite.

of 0.75°x0.75°. To reduce memory needs and maintain consistency with the CLaMS data approach, the CAMS data are also converted into monthly mean zonal mean data. To verify the impact of inter-annual differences, we compared the results between the simulations with CAMS reanalysis data for the years 2005 (matching the CLaMS data) and 2018 (matching the Ambae eruption), respectively. For another test, simulations with regional CAMS OH data extracted from the longitude range from 160°E to 140°W are compared with global data.

As shown in Fig. 8, the inter-annual differences in the CAMS OH data are negligible (∼1 percentage point, pp). The differences between simulations with CAMS and CLaMS global OH data are ∼3 pp. The simulation with regional CAMS OH data for the location of the eruption has ∼4 pp difference compared with the global CAMS data. Overall, the inter-annual and regional differences of the OH data sets have a small influence on the amount of OH oxidation (less than 5 pp in this case), which suggests that using a monthly mean zonal mean climatology is a reasonable approach for modeling the OH loss of $SO_2$ in the UT/LS region.



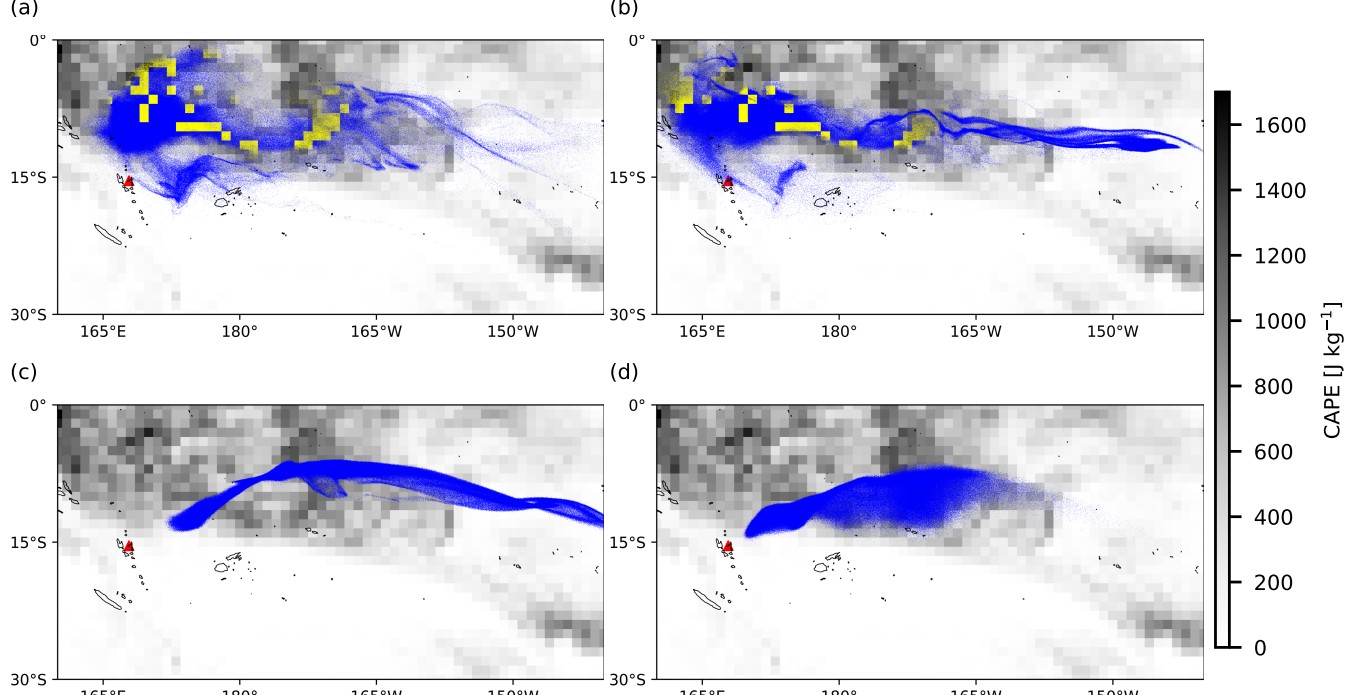

**Figure 7.** Same as Fig. 6, but background images represent the distribution of ERA5 CAPE values on 31 July 2018, 00:00 UTC. Yellow dots indicate air parcels below the equilibrium level and local CAPE values larger than $1000\,\mathrm{J\,kg^{-1}}$, triggering the convection module, while blue dots represent the opposite.

### 3.4 Sensitivity tests on wet deposition

In the Ambae case study, the wet removal in clouds plays an important role in depleting $SO_2$ since the main plume passes through a cloud region. As introduced in Sect. 2.1.3, in-cloud wet deposition is handled as a rain-out process, which depends

on the partition ratio of $SO_2$ in the cloud droplets and the precipitation rate. The partition ratio of $SO_2$ in liquid phase is defined with an effective Henry's law constant, depending on the assumed pH value. Here, we conducted a series of simulations to test the sensitivity to pH values in the range of 3 to 5, along with a comparison with the NAME wet deposition scheme (Webster and Thomson, 2014). As expected, higher cloud pH value lead to stronger wet deposition. The simulation with wet deposition according to the NAME scheme shows a similar deposition to the baseline simulation with the equilibrium scheme at pH 4 to

4.5. The difference between the two schemes is further reflected in the sensitivity with respect to convection, which will be discussed in Sect. 3.6

The retention ratio represents the solute species in the ice phase versus that in the liquid hydrometeor. The retention coefficient may vary depending on temperature, pH, ventilation rate, accretion rate, and impact velocity (Stuart and Jacobson, 2006). Laboratory studies on the retention ratio of $SO_2$ suggest a range from 1% to 60% (Iribarne et al., 1983; Lamb and Blumenstein,

1987; Iribarne et al., 1990). As introduced in Sect. 2.1.3, a retention coefficient dependent on temperature is applied to model





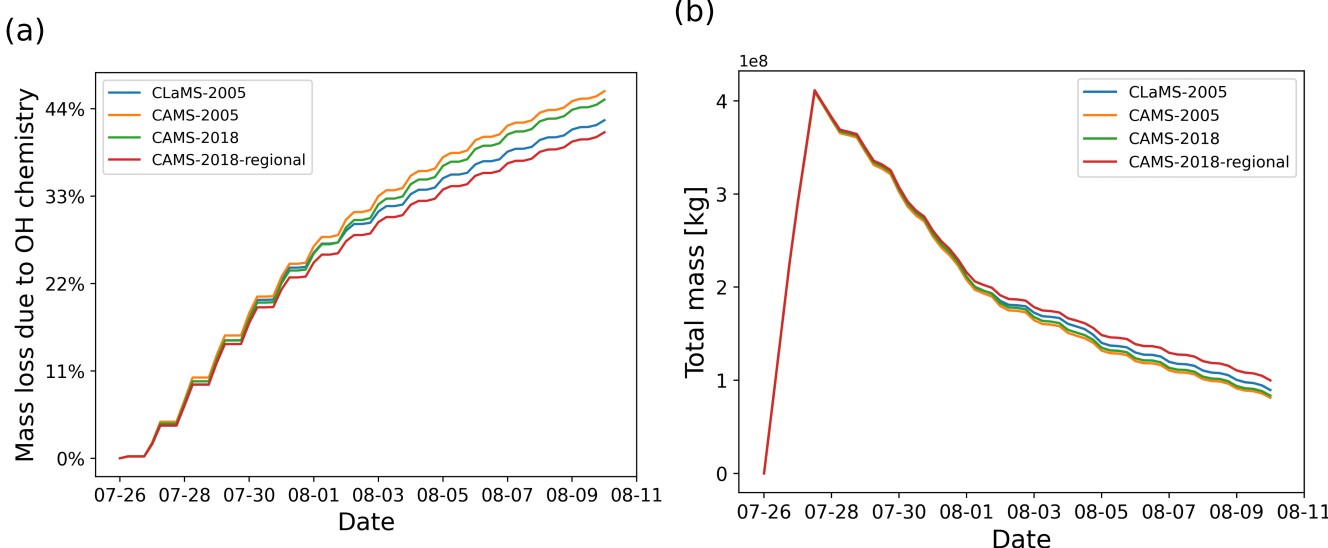

**Figure 8.** SO$_2$ mass loss with respect to total mass (in percent) due to the OH chemistry module (a) and total mass curves (b) for simulations with different OH datasets, including CLaMS data in 2005, CAMS data in 2005 and 2018, and CAMS regional data in 2018 (see text for details).

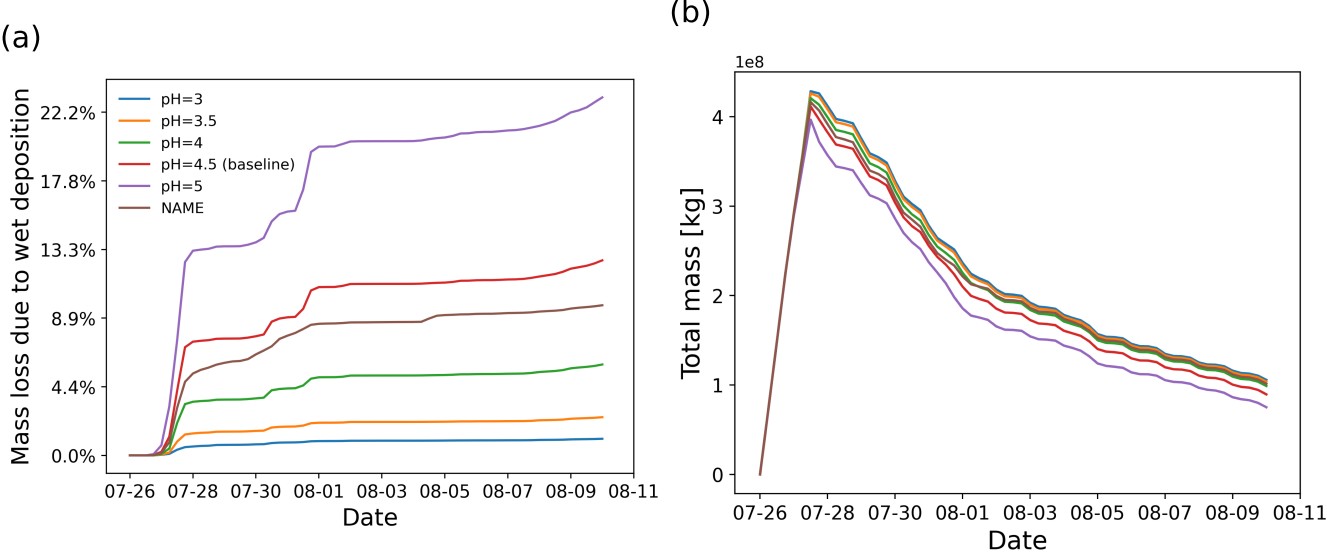

**Figure 9.** SO2 mass loss with respect to total mass (in percent) due the wet deposition module (a) and total mass curves (b), derived from simulations that use different pH values in the wet deposition module and comparing with the wet deposition scheme of the NAME model.




the differences of wet deposition in ice and liquid cloud. Figure 10 shows the $SO_2$ mass loss curves of simulations given different retention ratios in the range of 0.01 to 1. In this case, most of the wet deposition occurs in ice clouds leading to a strong dependency on the retention ratio.

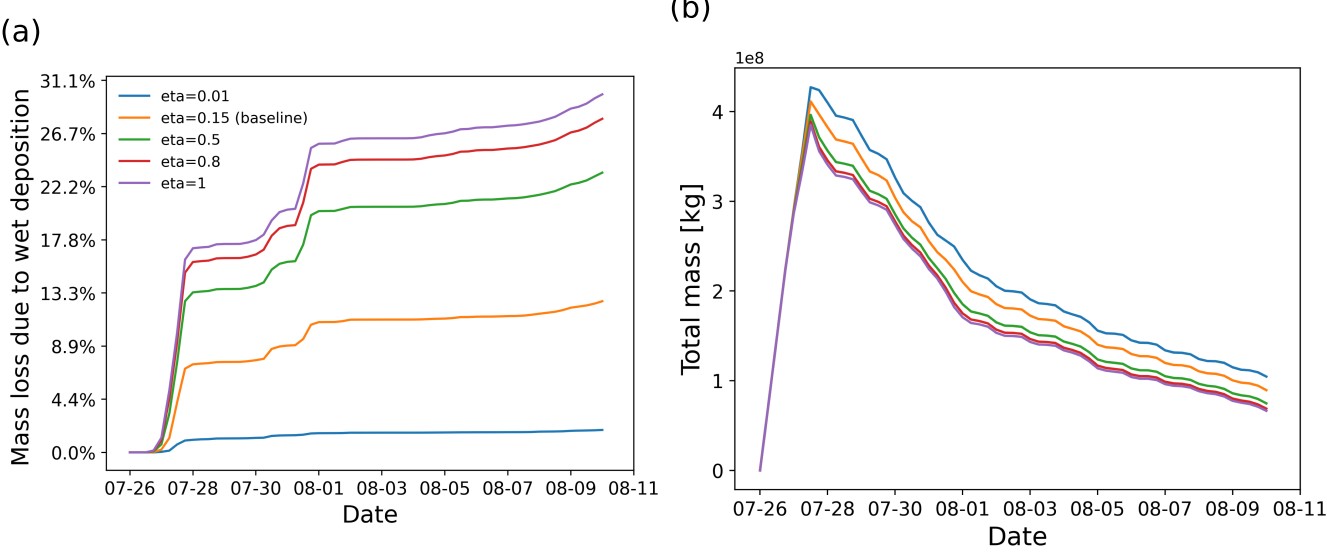

**Figure 10.** SO2 mass loss with respect to the total mass (in percent) due the wet deposition module(a) and total mass burden evolution (b), derived from different simulations that use different retention ratios settings in the wet deposition module.

### 3.5 Sensitivity tests on $H_2O_2$ chemistry

Since the meteorological data cannot fully resolve the convection and detailed cloud structures, a parameter cloud cover was defined in the $H_2O_2$ chemistry module to represent the probability of an air parcel to be located in a cloud. When a random number in the range of 0 to 1 is larger than a given threshold for the cloud cover, the $H_2O_2$ decomposition will be skipped. Figure 11 shows the dependency of aqueous phase $H_2O_2$ oxidation on cloud cover. Compared to full cloud cover, the amount of $H_2O_2$ oxidation in a case with only 10% cloud cover is decreased by 8 pp. However, the $SO_2$ total mass burden is only 365  slightly change.

### 3.6 Impact of the extreme convection parametrization

Due to better spatiotemporal resolution, the ERA5 meteorological data provide more accurate information than ERA-Interim to resolve mesoscale features (Hoffmann et al., 2019). However, convective up- and downdrafts are still underrepresented in the ERA5 data. In MPTRAC, the extreme convection parametrization is applied to represent the effects of unresolved convection in 370  the meteorological input data. In convective columns, the convection module will randomly redistribute the air parcels between the surface and the equilibrium level, if CAPE exceeds a given threshold $CAPE_0$. As shown in Fig. 7, the $SO_2$ plume released



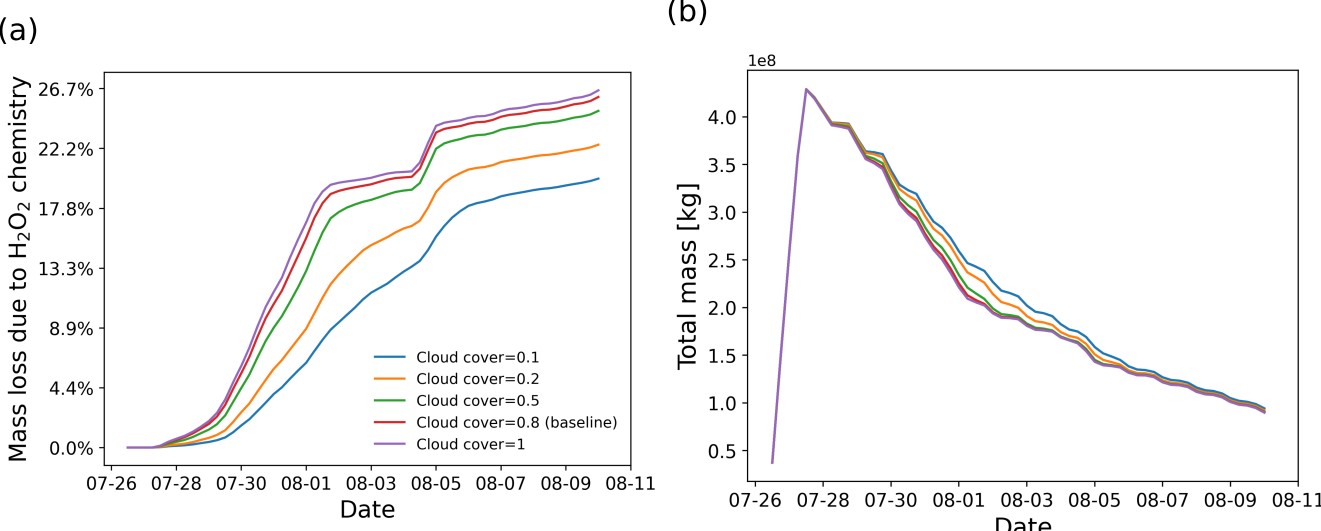

**Figure 11.** SO2 mass loss with respect to total mass (in percent) due the $H_2O_2$ chemistry module (a) and total mass burden evolution (b), derived from simulations that use different cloud cover settings in the $H_2O_2$ chemistry module.

by the Ambae volcanic eruption was transported to a high CAPE region which makes it a valuable case to study the potential effects of unresolved convection. In this section, the effects of parametrized convection for different thresholds $CAPE_0$ will be discussed.

Figure 12a shows the vertical distributions of the $SO_2$ mass of the Ambae plume for different thresholds $CAPE_0$ from 0 to $2000\,J\,kg^{-1}$. The case of $CAPE_0$ being zero represents the extreme case in which convection occurs everywhere below the equilibrium level where CAPE is present. For $CAPE_0$ larger than $1000\,J\,kg^{-1}$, the parametrized convection is restricted to represent moderate to strong convective events only. There is almost no additional convection for $CAPE_0$ larger than $2000\,J\,kg^{-1}$ compared to the case without parametrized convection. When applying a larger $CAPE_0$, more air parcels will be transported to

altitudes below the equilibrium level ($\sim$14 km).

Figure 13 shows the impact of parametrized convection on mass loss in the $SO_2$ transport simulations for the Ambae case. The largest sensitivity is found for the $H_2O_2$ oxidation, while OH oxidation and wet deposition show little difference. This is because some air parcels are transported to heights below 14 km where the $H_2O_2$ oxidation is most active. The $H_2O_2$ chemistry is much more sensitive to height than the OH chemistry and wet deposition at altitudes where convection occurs (see Sect. 3.2).

Regarding the mass evolution, the effect of the convection module enhances the depletion of $SO_2$ below the cloud top. At the end of simulation, most of the remaining $SO_2$ is above cloud top where the convection module takes no effect.



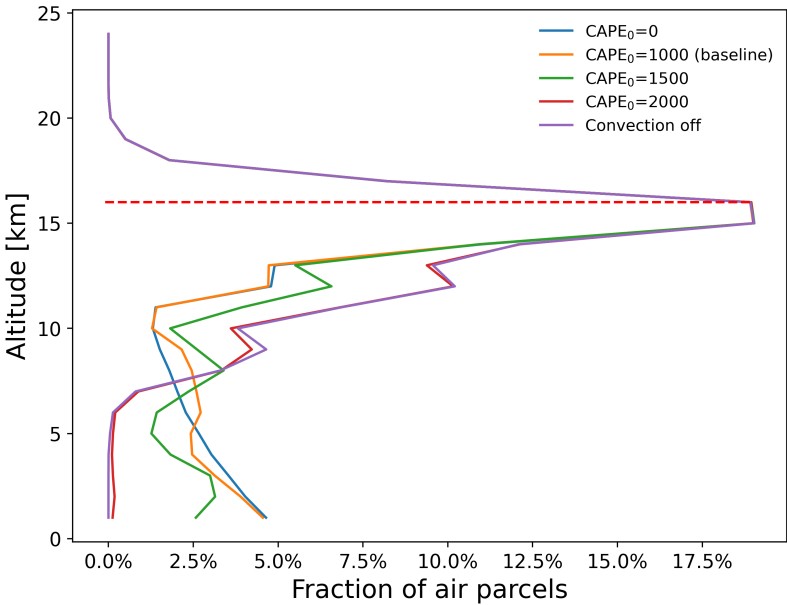

**Figure 12.** Vertical distribution of total $SO_2$ mass of the air parcels on 1 August 2018, 00:00 UTC from simulations with different CAPE thresholds. The red dashed line represents the height of the tropopause.

## 4 Discussion and conclusions

We present recent improvements of the chemical and physical modules in the Lagrangian transport model MPTRAC for simulating the dispersion and depletion of $SO_2$ from volcanic eruptions. The improved modules include gas phase OH oxidation, aqueous phase $H_2O_2$ oxidation, wet deposition, and parametrized convection. In a case study, the modules are applied for simulations of the $SO_2$ plume evolution of the Ambae eruption in July 2018. We present a baseline simulation, in which the modeled mass evolution shows a good match with TROPOMI satellite observations.

In our simulations, the implementation of wet deposition and $H_2O_2$ oxidation significantly decreased the lifetime of $SO_2$ in clouds. Gas phase oxidation by OH, aqueous phase oxidation by $H_2O_2$, and wet deposition remove about 46%, 14%, and 26% of the $SO_2$ total mass burden during a simulation time period of 15 days, respectively. Based on the baseline simulation, a series of sensitivity tests was conducted by tuning various control parameters of the modules to better understand the impacts of the chemical and physical processes on $SO_2$ dispersion. Initially, we tested the sensitivity on the assumed injection height of the $SO_2$ plume. The $SO_2$ plume undergoes much faster depletion below 14 km in clouds than in the dry atmosphere because rainout processes and aqueous phase oxidation are bound to the presence of clouds. Variations of cloud distributions and vertical wind shear lead to rather different amount of wet deposition and aqueous phase oxidation for different injection heights.

For the OH chemistry module, different OH monthly mean zonal mean concentration data sets have been tested. It was found that the differences in mass loss between different OH datasets (CAMS versus CLaMS, regional versus global, and year





**Figure 13.** Same as Fig. 5, but derived from simulations with different CAPE thresholds in the convection module.

2005 versus 2018) are limited (less than 5 pp of the total mass). Using a monthly mean zonal mean climatology of OH for simulating $SO_2$ mass loss over time is considered a suitable approach for transport simulations covering the UT/LS region.

For wet deposition, an analytic solution derived for the rain-out process according to Henry's law is considered in cloud. The assumed pH value of the effective Henry's Law constant for $SO_2$ has a strong effect on the amount of wet removal of $SO_2$. Compared with the wet deposition scheme with a scavenging coefficient formula applied in the UK Met Office's NAME model, our simulation at an assumed pH value of 4.5 yields a similar level of wet deposition. The impact of the retention ratio of the soluble gas in ice cloud is also discussed. Due to the low temperatures in the UT/LS region, most of the regions affected by

the Ambae eruption are covered with ice clouds. Therefore, the amount of wet deposition strongly depends on the assumed retention ratio. For $H_2O_2$ chemistry in the aqueous phase, the impact of cloud cover has been tested. By tuning the CAPE



threshold used in the extreme convection parametrization, the vertical distribution of the $SO_2$ plume might be strongly altered, which in turn will impact the depletion of $SO_2$. It is found that aqueous phase oxidation via $H_2O_2$ is mostly affected, whereas wet deposition and OH oxidation are not so sensitive to the vertical redistribution of the air parcels due to the parametrized convection.

In this study, we improved the Lagrangian transport simulations of volcanic $SO_2$ with MPTRAC by implementing more realistic physical and chemical process representations instead of using an exponential decay law with a fixed lifetime as in our earlier studies. However, the refinements on the modeled loss processes lead to a high sensitivity to the release vertical profile and cloud abundance. The backward trajectory method developed to estimate time- and height-resolved $SO_2$ emissions with MPTRAC (Hoffmann et al., 2016; Wu et al., 2017) is currently incapable of taking into account the additional complexity of the revised chemistry and wet deposition parametrizations. In future work, we plan to implement a more advanced inverse modeling techniques for MPTRAC to reconstruct the initial $SO_2$ injections of volcanic eruptions, which can take the model improvements described here into account.

*Code and data availability.* MPTRAC (Hoffmann et al., 2016, 2022a) is made available under the terms and conditions of the GNU General Public License (GPL) version 3. The release version 2.4 of MPTRAC applied in this paper has been archived on Zenodo (Hoffmann et al., 2022b). New versions of MPTRAC are made available via the repository at https://github.com/slcs-jsc/mptrac (last access: 20 December 2022). The ERA5 reanalysis (Hersbach et al., 2020) was retrieved from ECMWF's Meteorological Archival and Retrieval System (MARS). See https://www.ecmwf.int/en/forecasts/datasets/browse-reanalysis-datasets (last access: 20 December 2022) for further details.

## Appendix A: Validation of OH fields with diurnal variability

A monthly mean zonal mean OH climatology with diurnal corrections according to Eq. (6) is used as input to calculate the decomposition of $SO_2$ due to OH oxidation. For validation of the OH data, we extracted OH values from MPTRAC as a function of the solar zenith angle and compared them with in-situ measurement data provided by the NASA Earth Science Project Office (ESPO, 2022). Here, we present comparisons for April (Fig. A1) and October (Fig. A2) containing in-situ measurements of campaigns SUCCESS-DC8, SONEX-DC8, POLARIS-ER2, and MAESA-ER2, covering altitudes ranging from 5 to 21 km. For the stratosphere, a comparison with OH data from MLS is shown in Fig A3. It is found that CLaMS OH data better represent the measured OH concentrations than the CAMS reanalysis in the stratosphere. For the troposphere, the OH data from CLaMS and CAMS both match the measurements well.

*Author contributions.* ML, LH, and SG jointly developed the concept of this study. ML conducted most of the model development, performed the model simulations, and analysed the results. LH contributed to MPTRAC model development. SG contributed expertise on OH measurements. ZC, SG, LH, YH, and XW provided expertise on Lagrangian transport modelling for volcanic eruptions. ML wrote the manuscript with contributions from all co-authors.



**Figure A1.** OH volume mixing ratios used for MPTRAC based on predefined monthly mean zonal mean data for the CLaMS model (red curve) and the CAMS reanalysis (blue curve) as well as OH in-situ measurements from the campaigns SUCCESS-DC8 (black points) and POLARIS-ER2 (green points). The time period being covered is the month of April. Plots in (a) to (i) refer to altitudes of 5, 7, 9, 11, 13, 15, 17, 19, and 21 km, respectively.







**Figure A2.** Same as Fig. A1 but for SONEX-DC8 (black points) and MAESA-ER2 (green points) measurements in October.

*Competing interests.* The authors declare that no competing interests are present.

*Acknowledgements.* The work described in this paper was supported by the Helmholtz Association of German Research Centres (HGF) through the Joint Lab Exascale Earth System Modelling (JL-ExaESM). We acknowledge the Jülich Supercomputing Centre for providing computing time and storage resources on the supercomputer JUWELS. We acknowledge Sandra Wallis for providing TROPOMI SO₂ mass data at an earlier stage of this study. Mingzhao Liu acknowledges the support provided by the China Scholarship Council (CSC Grant no. 202006380039). Xue Wu is supported by the National Natural Science Foundation of China (Grant 41975049) and the Basic Strengthening Research Program (Grant 2021-JCJQ-JJ-1058). Yi Heng acknowledges the support provided by Key-Area Research and Development Program of Guangdong Province (No.2021B0101190003), Zhujiang Talent Program of Guangdong Province (No.2017GC010576).





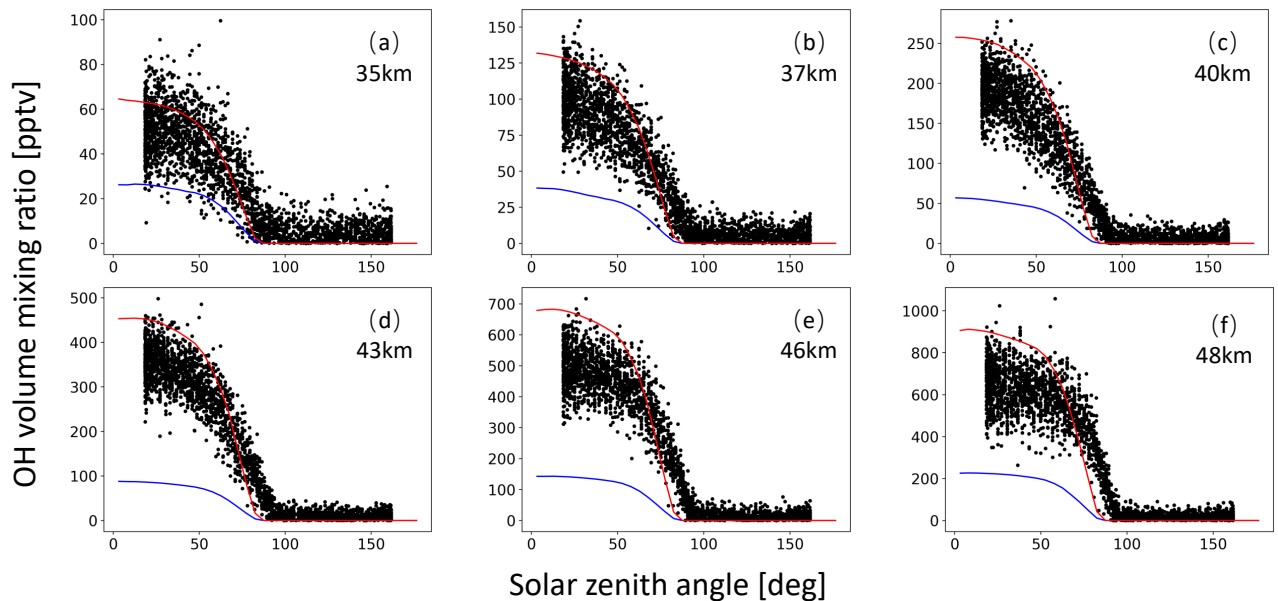

**Figure A3.** Global average OH volume mixing ratios used in MPTRAC based on predefined monthly mean zonal mean data for the CLaMS model (red curve) and the CAMS reanalysis (blue curve) as well as MLS satellite measurements (black points). The time period being covered is May. Plots in (a) to (f) refer to altitudes of 35, 37, 40, 43, 46, and 48 km, respectively.

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
