# Peer review of "Improved representation of volcanic sulfur dioxide depletion in Lagrangian transport simulations: a case study with MPTRAC v2.4"

_EGUsphere, 2022_

## Author Comment (AC1)

**Reply to review comments**

We thank the reviewers and the editor for their time and effort in reviewing the manuscript and for their helpful comments. We have considered all the comments and hope that the revised draft adequately addresses the open issues. Please find below our point-by-point responses (colored in blue). A revised manuscript with tracked changes has been uploaded.

**Reviewer #1**

General comments:

The paper presents new developments to include new or improved $SO_2$ removal processes in a Lagrangian transport model. Accurate representation of the emissions, transport and removal of atmospheric constituents is crucial for accurate modelling and forecasting, and therefore the paper is highly relevant for the field of atmospheric modelling. The paper shows a breakdown of the contribution of different removal processes for $SO_2$ (as a % of total mass removed), similar to the Berglen 2004 paper, but with more details and focus on $SO_2$. Such details have been lacking in the literature. The paper is very well written, figures are exceptional and extensive sensitivity tests are presented in a very clear and concise way.

We would like to thank the reviewer for the encouraging statement.

The paper is highly suited for publications with only a few minor comments to consider, as specified below:

Main comments:

The comparison between the modelled $SO_2$ mass to the TROPOMI observations in Figure 3a shows a very good agreement. The comparison is done on the TROPOMI horizontal footprint and times. However, such a comparison also need to consider the Averaging Kernel (AK) which characterize the vertical sensitivity of the measurements, and which is required for comparison with other types of data. This AK should be applied to the model data before comparing to the observations (as in Kristiansen et al 2010 doi:10.1029/2009JD013286). As the $SO_2$ emissions and plume height for the case considered (Ambae) is mostly > 10 km, I do not think this will change the results significantly, however it should still be included. The AK is available as an output field of the TROPOMI retrieval.

Thanks for bringing this issue to our attention. We agree that the vertical sensitivity of the TROPOMI satellite measurements should be taken into account when comparing the model results with the observations. As suggested, we applied the averaging kernel from the TROPOMI retrieval to the model results and revised the manuscript accordingly. The averaging kernel does not change the results significantly, as shown in Fig. 1 in this

[Figure]

Figure 1: Observed (TROPOMI) and simulated (MPTRAC) SO₂ total mass curve of the July 2018 Ambae eruption as a function of time. The blue shading represents the total error of the TROPOMI observations, combining random and systematic error components.

document and as expected by the reviewer. We have added such statements in the revised manuscript: "The model results are multiplied by an altitude-dependent sensitivity profile, which was derived from TROPOMI averaging kernel data. The TROPOMI averaging kernel indicates full sensitivity above 10 km of altitude and the sensitivity only significantly decreases in the lower troposphere Cai et al. (2022). Since in this case most of the SO₂ injections and the plume height were located over 10 km, considering the averaging kernel does not change the model results significantly."

The comparisons to the SO₂ removal processes in the FLEXPART and NAME models could be made clearer. In particular, the Introduction could include an explicit statement that the new developments in MPTRAC are compared to SO₂ removal schemes in similar Lagrangian models with a short explanation of the removal schemes in other models. Also acknowledging that many other types of models (e.g., Eulerian CTMs) include quite complex chemistry schemes. Furthermore, the wet deposition scheme in FLEXPART is mentioned, but the SO₂ chemistry in FLEXPART is not and should be highlighted (Eckhardt et al 2008 https://doi.org/10.5194/acp-8-3881-2008). Also, both FLEXPART and NAME use OH climatological background fields in their SO₂ chemistry schemes and this should be mentioned as its highly relevant to your findings.

We extended the Introduction to reference other Lagrangian modelling studies on volcanic SO₂ using the FLEXPART and NAME models and to relate them to the current scheme applied in MPTRAC. Since this study focuses on Lagrangian modelling, we have not mentioned other types of models (e.g., Eulerian CTMs) with complex chemistry schemes to

avoid distraction. We added "Various studies applied Lagrangian particle dispersion models for simulations of volcanic $SO_2$ plume transport. The study by Eckhardt et al. (2008) with the Flexible Particle (FLEXPART) model considered removal of $SO_2$ by reaction with OH radicals while aqueous-phase chemistry reactions were not considered. The study by de Leeuw et al. (2021) with the Met Office's Numerical Atmospheric dispersion Modelling Environment (NAME) model considered the conversion of $SO_2$ into sulfate aerosol with both gas phase and aqueous phase oxidation. In this work, we aim to model the removal of $SO_2$ from volcanic plumes with a more realistic representation of physical and chemical processes in the MPTRAC model, including gas and aqueous phase oxidation as well as wet deposition. The chemical loss is represented by first-order rate coefficients derived from predefined, climatological OH and $H_2O_2$ fields. Chemistry calculations are conducted in the Lagrangian framework rather than using a Eulerian framework, which avoids memory sharing and is well suited for parallel processing. The approach achieves a balance between computational costs and accuracy, which is similar to the FLEXPART model but different from the full chemistry scheme applied in NAME."

It's not entirely clear how the chemistry calculations are performed in MPTRAC. Are they done in Lagrangian or Eulerian (i.e., gridded) space? Please provide some more details.

The chemistry calculations are done in Lagrangian space. OH and $H_2O_2$ oxidation loss are considered but no chemical production. The oxidation rate is prescribed first-order with predefined OH and $H_2O_2$ fields. The air parcel mass is reduced by an exponential decay according to the rate coefficient. This simplified chemistry scheme can represent the volcanic $SO_2$ lifetime discrepancies at different atmospheric conditions above/below clouds. As suggested, we added more details in the Introduction to better explain this approach.

Please comment on whether the diurnal variability in OH is a significant factor. It gives a stepwise appearance on the plots, but the importance of this is not much discussed.

Figure 2 in this reply shows the comparison of the OH loss curve with and without correction for diurnal variations. On average, the effect of the diurnal variability of OH is not very significant over the whole simulation period. Effects appear only when focusing on specific local times, such as sunrise or sunset. We added more discussion on the effects of the OH diurnal variations to the manuscript, "The loss due to OH oxidation is a step-shaped curve due to the diurnal variations of the OH concentration, leading to a faster loss rate at daytime and nearly zero loss at nighttime. Simulations using an OH field with diurnal variations will vary by 0.7% compared to using an OH field without diurnal variations. Differences are relatively low in this case because the observations occur at about 13:00 local time. Overall, the diurnal variations of the OH field have little effect on the $SO_2$ decay rates over the entire simulation period, because the observational time scale is over several days and the diurnal variations are averaging out."

The applicability/suitability of an atmospheric $SO_2$ chemistry scheme to volcanic clouds could be discussed in more general. Often these schemes are developed for air quality purposes, which may involve different atmospheric species than those present in volcanic

[Figure]

Figure 2: Simulated (MPTRAC) OH loss curves with and without diurnal variations of OH as a function of time.

clouds, and also with a focus on the lower atmosphere. Volcanic eruptions that release significant amounts of water to the atmosphere (e.g., the HTHH 2021 eruption) could have a different chemistry in the plume, and the additional water/ice will likely not be included in driving meteorological fields and will need to be treated separately.

In this paper, we aim to better represent the $SO_2$ lifetime for volcanic eruptions by considering first-order loss processes but not by implementing more detailed chemistry schemes. In specific volcanic eruption cases, additional processes or more complex representations need to be involved, such as absorption by volcanic ash or reaction with strong release of water vapor. However, to take these factors into account, the mechanism of the dynamics within these phenomena needs to be further investigated. Further simulation case studies are needed to test the applicability of the $SO_2$ decay representation presented in this paper to other volcanic eruptions in the future. We are currently developing a full chemistry scheme with an Eulerian approach for MPTRAC, in order to represent more complicated reaction schemes and to consider more species, which has the potential to be used in future studies. We added an outlook in the Conclusions, "The representation of volcanic $SO_2$ depletion in different atmospheric conditions above and below the cloud top in the current MPTRAC scheme considers first-order loss processes to estimate rate coefficients and lifetimes. In reality, the chemical processes affecting the lifetime of volcanic $SO_2$ may be more complicated. Large-scale emissions of volcanic $SO_2$ into the atmosphere will reduce the abundance of atmospheric oxides. The scattering and reduction of solar radiation by ash and other aerosol particles will reduce the photochemical regeneration of the oxides. Other factors, such as additional water release from an eruption or absorption by volcanic ash, need to be further investigated to better understand and elucidate the mechanism of their dynamics. In future work, we intend to further test and improve the current version

of MPTRAC to explore its applicability to other volcanic eruption cases."

Specific comments

L51: Comparing the total $SO_2$ emitted from the Ambae 2018 eruption to that released from the other eruptions mentioned (Raikoke, Sarychev, Nabro, Kasatochi) would give some perspective on the size of the $SO_2$ emissions.

As suggested, in the Introduction we referenced and added the release amount of $SO_2$ of the other eruptions mentioned in the paper. These eruptions had larger releases of about 1500, 2000, 1200, and 3650 kt, respectively.

Technical comments

L365: The word 'change' should be 'changed'.

L375: Figure 12a should be Figure 12.

We fixed the technical comments as suggested.

**Reviewer #2**

General comments:

This paper introduces the implementation of several new chemical and physical $SO_2$ removal schemes into the Massive-Parallel Trajectory Calculations (MPTRAC) Lagrangian transport model. The paper demonstrates how different processes contribute to the depletion of volcanic $SO_2$ in the atmosphere. Using the 2018 Ambae eruption as case study, the authors compare the simulated $SO_2$ lifetime with the observed $SO_2$ lifetimes from the Atmospheric InfraRed Sounder (AIRS) and the TROPOspheric Monitoring Instrument (TROPOMI). By performing multiple sensitivity tests on the new schemes through adjustment of the tuning parameters, it is found that the modelled $SO_2$ loss processes are highly sensitive to the release vertical profile and the presence of clouds.

Overall, the paper provides an useful evaluation of the new chemical and physical schemes in the MPTRAC model, and the topic of this paper will be of interest to the readers of this journal. Accurate representations of removal processes of atmospheric constituents are highly relevant for the skill of a transport model and its forecasting abilities. Therefore, this work is relevant for the wider atmospheric modelling community.

The paper is well structured, the quality of the figures is good, and the authors give a sufficient introduction to the new schemes and a clear and concise interpretation of the data and sensitivity studies.

We would like to thank the reviewer for the positive assessment of the manuscript.

I therefore recommend minor revisions to address the points outlined below before publication.

Main comments:

When comparing the model results with the TROPOMI satellite retrievals in section 3.1, the model simulations should be adjusted using the Averaging Kernel (AK). As mentioned in the methodology section (L.244), the TROPOMI product assumes the $SO_2$ at a particular altitude in the atmosphere. This also results in a specific vertical sensitivity of the satellite $SO_2$ retrievals, where the satellite is in general more sensitive to higher altitudes. No such sensitivities are present in the model, and therefore one needs to apply the same vertical sensitivity profile from the satellite retrievals to the model data (i.e. the AK) to make the comparison correct. As most of the $SO_2$ is above 10km, the impact will most likely be small, but it should be included.

We revised the paper to include the TROPOMI averaging kernel when comparing simulation results and observations. The overall effect of this is small, as expected. Please see reply to Reviewer #1 for further details.

In the introduction, I think the paper would benefit from a short discussion of the relevant $SO_2$ removal schemes present in other Lagrangian transport models (e.g. NAME and FLEXPART) to help to put your work in context with these existing schemes.

We extended the Introduction of the paper accordingly, also following the comments provided by both reviewers.

Specific comments:

L.69: How are the chemistry calculations done in the MPTRAC model? A short discussion of the implementation of the schemes into the model is missing here.

Please see reply to respective comment by Reviewer #1.

L.116: The implementation of the diurnal variability is mentioned here, but no discussion of the impact on the results in presented in the manuscript. In several figures (e.g. figures 3b, 5b and figure 8) the impact is visible, but it is not clear to me if this is actually having any significant impact on the overall depletion of $SO_2$. Is it essential to include the diurnal variation, or could this be ignored in future simulations?

Please see reply to respective comment by Reviewer #1.

Figure 2: Due to the large number of grey points, it is not possible to understand if the two regression lines are a good representation of the data. I think it would be better if the data could be shown as a density plot.

Following the comment, we added color coding to the scatter plot to show the normalized density of the data points, please see Fig. 3 in this reply and the revised manuscript.

Figure 12: How is the tropopause height calculated?

The tropopause height was calculated according to the standard WMO lapse rate criterion

[Figure]

Figure 3: Scatter density plot of total column cloud water versus precipitation rate of ERA5 meteorological data for the region and time period covering the Ambae eruption. The black line represents a regression model of the data. The red line shows the expression given by Pisso et al. (2019).

(WMO, 1957). For details of the calculation and for the tropopause data, we added a reference to Hoffmann and Spang (2022).

Technical corrections/suggestions:

L.14: physical of $SO_2$ -> physical $SO_2$

L.109: climatology(Inness et al., 2019) -> space is missing

L.365: change -> changed

L.375: Figure 12a -> Figure 12

Thank you for pointing out the technical corrections. We fixed the text as suggested.

**References**

Cai, Z., Griessbach, S., and Hoffmann, L.: Improved estimation of volcanic $SO_2$ injections from satellite retrievals and Lagrangian transport simulations: the 2019 Raikoke eruption, Atmos. Chem. Phys., 22, 6787–6809, doi: 10.5194/acp-22-6787-2022, 2022.

de Leeuw, J., Schmidt, A., Witham, C. S., Theys, N., Taylor, I. A., Grainger, R. G., Pope, R. J., Haywood, J., Osborne, M., and Kristiansen, N. I.: The 2019 Raikoke volcanic

eruption – Part 1: Dispersion model simulations and satellite retrievals of volcanic sulfur dioxide, Atmos. Chem. Phys., 21, 10 851–10 879, doi: 10.5194/acp-21-10851-2021, 2021.

Eckhardt, S., Prata, A. J., Seibert, P., Stebel, K., and Stohl, A.: Estimation of the vertical profile of sulfur dioxide injection into the atmosphere by a volcanic eruption using satellite column measurements and inverse transport modeling, Atmos. Chem. Phys., 8, 3881–3897, doi: 10.5194/acp-8-3881-2008, 2008.

Hoffmann, L. and Spang, R.: An assessment of tropopause characteristics of the ERA5 and ERA-Interim meteorological reanalyses, Atmos. Chem. Phys., 22, 4019–4046, doi: 10.5194/acp-22-4019-2022, 2022.

Pisso, I., Sollum, E., Grythe, H., Kristiansen, I, N., Cassiani, M., Eckhardt, S., Arnold, D., Morton, D., Thompson, R. L., Zwaaftink, C. D. G., Evangeliou, N., Sodemann, H., Haimberger, L., Henne, S., Brunner, D., Burkhart, J. F., Fouilloux, A., Brioude, J., Philipp, A., Seibert, P., and Stohl, A.: The Lagrangian particle dispersion model FLEXPART version 10.4, Geosci. Model Dev., 12, 4955–4997, doi: 10.5194/gmd-12-4955-2019, 2019.

WMO: Meteorology A Three-Dimensional Science: Second Session of the Commission for Aerology, WMO Bull., iv, 134–138, 1957.

---

## Author Response (AR2)

**Reply to editor comments**

Dear editor,

thank you for the time and effort spent on the manuscript. We considered the initial comments and hope that the revised draft properly addresses the open issues. Please find our point-by-point replies below (colored in blue). A revised manuscript with tracked changes has been uploaded.

Best regards,

Mingzhao Liu

**Editor's comments**

Dear authors,

you addressed all points raised by the reviewer and made according changes in the manuscript. Before I can accept it for publication in GMD, two issues have to be resolved.

1.) The data availability section needs to point to a second (Zenodo) repository, where all scripts that are required to reproduce the study are collected. Configuration files of simulations runs and plotting scripts. Please have a look at existing GMD publications if it is no fully clear what is meant (or contact me directly).

The scripts to reproduce this study has been uploaded on Zenodo (Liu, 2023) and cited in the code availability of paper.

2.) There is a difference between "which" and "that" in English language

(https://www.diffen.com/difference/That_vs_Which).

I spotted several sentences where the wrong word was used. E.g. "MPTRAC is a Lagrangian transport model, which " needs 'that' Similarly, "by an altitude-dependent sensitivity profile, which" needs 'that' Please go over the manuscript to correct such mistakes.

Thank you for point out the mistakes. We have check over the manuscript to make some revisions on such wrong use of word.

Best wishes,

Simon Unterstrasser

**References**

Liu, M.: Running script of modeling the Ambae eruption SO2 transport in July 2018 with MPTRAC v2.4, doi: 10.5281/zenodo.8072111, 2023.

---

## Author Response (AR3)

**Reply to editor comments**

Dear editor,

Thank you for your suggestions. We added the README file with detailed descriptions and some additional plot scripts in the updated version of Zenodo repository (Liu et al., 2023).

Best regards,

Mingzhao Liu

**References**

Liu, M., Hoffmann, L., and Griessbach, S.: Running script of modeling the Ambae eruption SO2 transport in July 2018 with MPTRAC v2.4, doi: 10.5281/zenodo.8163071, 2023.